# How to Monitor Hydration Status and Urine Dilution in Patients with Nephrolithiasis

**DOI:** 10.3390/nu15071642

**Published:** 2023-03-28

**Authors:** Simon Travers, Caroline Prot-Bertoye, Michel Daudon, Marie Courbebaisse, Stéphanie Baron

**Affiliations:** 1Université Paris Cité, Paris Cardiovascular Research Center, Inserm U970, F-75015 Paris, France; 2Service de Physiologie, Assistance Publique-Hôpitaux de Paris, Hôpital Européen Georges Pompidou, F-75015 Paris, France; 3Centre de Référence des Maladies Rénales Héréditaires de l’Enfant et de l’Adulte (MARHEA), F-75015 Paris, France; 4Centre de Référence des Maladies Rares du Calcium et du Phosphate, F-75015 Paris, France; 5Université Paris Cité, Sorbonne Université, Centre de Recherche des Cordeliers, Inserm U1138, F-75006 Paris, France; 6Centre National pour la Recherche Scientifique, EMR 8228, Laboratoire de Physiologie Rénale et Tubulopathies, F-75006 Paris, France; 7Sorbonne Université, Inserm U1155, Hôpital Tenon, F-75020 Paris, France; 8Université Paris Cité, Inserm U1151, Institut Necker Enfants Malades, F-75015 Paris, France

**Keywords:** kidney stone prevention, kidney stone recurrence, osmolality, urine-specific gravity

## Abstract

Maintenance of hydration status requires a tight balance between fluid input and output. An increase in water loss or a decrease in fluid intake is responsible for dehydration status, leading to kidney water reabsorption. Thus, urine volume decreases and concentration of the different solutes increases. Urine dilution is the main recommendation to prevent kidney stone recurrence. Monitoring hydration status and urine dilution is key to preventing stone recurrence. This monitoring could either be performed via spot urine or 24 h urine collection with corresponding interpretation criteria. In laboratory conditions, urine osmolality measurement is the best tool to evaluate urine dilution, with less interference than urine-specific gravity measurement. However, this evaluation is only available during time lab examination. To improve urine dilution in nephrolithiasis patients in daily life, such monitoring should also be available at home. Urine color is of poor interest, but reagent strips with urine-specific gravity estimation are currently the only available tool, even with well-known interferences. Finally, at home, fluid intake monitoring could be an alternative to urine dilution monitoring. Eventually, the use of a connected device seems to be the most promising solution.

## 1. Introduction

### 1.1. Regulation of Hydration Balance

Maintenance of hydration status requires a tight balance between fluid output and input. This balance should yield the maintenance of euhydration. Nonetheless, in numerous situations, excess loss of water or insufficient fluid intake could induce a state of water depletion. This dehydration status could be acute (because of a punctual situation, such as diarrhea) or chronic, and it could range from moderate to even severe dehydration status. This hydration imbalance is the consequence of excess water loss and/or a deficiency in fluid input.

Because only a small amount of water is produced by our metabolism (around 250 mL/day), water intake is the most important part of fluid input. Most human fluid intake comes from pure water (around 61% of total daily water intake), while the water we ingest in the form of other beverages or water contained in food represents less than 40% of total daily water intake [1]. Regarding fluid output, loss of water mainly occurs via kidney urine excretion, via sweating, through the respiratory tract and via the feces, as described in Figure 1. Under normal conditions, extrarenal water loss is low and the volume of urine is equal to the volume of fluid intake. Only water excretion by the kidneys can be regulated, while the other three pathways of water loss (sweat, respiratory and feces) are not. Kidneys control urine volume to maintain water balance. In an 18° to 25 °C environment, a healthy sedentary adult will have moderate water loss ranging from 1.8 L/day to 3.0 L/day [2]. Of note, environmental temperature, altitude, humidity level, physical activity and diet can also affect water loss [3,4]. In the same way, seasonal variations could be responsible for water depletion, particularly among some susceptible subpopulations, such as the elderly [5,6], particularly fragile patients, i.e., patients on antipsychotics [7], and patients suffering from cystic fibrosis [8]. This is also the case for manual workers working under extreme heat conditions [9,10].

### 1.2. Water Intake and Regulatory Mechanisms

Dehydration is hypertonic when water loss exceeds electrolyte loss, leading to a higher electrolyte blood concentration and thus to an increase in plasma osmolality, reflecting an intracellular dehydration status [11,12,13]. To correct this intracellular dehydration, a water shift from the extra to the intracellular compartment occurs (Figure 1). Increase in plasma osmolality is the main factor stimulating the two homeostatic mechanisms; first, antidiuretic hormone arginine vasopressin (AVP) release occurs, followed by thirst stimulation. AVP is a peptide synthesized in the supraoptic and paraventricular nuclei of the hypothalamus and is released from the posterior pituitary [14]. AVP activates the V2-receptor in the renal distal tubule, leading to an increase in the production of water channels (aquaporin) and their insertion into the luminal membrane. This promotes water reabsorption from tubular fluid to blood, rendering the tubular fluid more concentrated [15,16]. The capacity of the kidney to excrete a concentrated urine is the first defense mechanism against water depletion. This mechanism leads to an increase in urine osmolality thanks to the concentration capacity of the kidney. When urine water reabsorption is insufficient to maintain plasma osmolality (>300 mOsm/kg H_2_O), thirst is triggered to increase water intake. Indeed, an increase of 1 or 2 percent in plasma osmolality elicits thirst and fluid intake. Nonetheless, perception of thirst, especially in the elderly (over 65 years), could be altered [17,18,19], or thirst could be neglected [20]. Thus, a large part of the population finally exhibits a moderate dehydration status. In 14,855 American community-dwelling adults (20–90 years) who gave blood for the third National Health and Nutrition Examination Survey, 60% of these adults presented with measured hypertonic plasma [21]. Even with ad libitum access to fluid, a recent German population-based observational study showed that median total water intake decreased with increasing age, only in males [22]. Finally, any acute or chronic dehydration, even moderate or mild, is responsible for an increase in kidney water reabsorption, a decrease in urine volume and an increase in different solutes’ concentration (see Figure 1, red arrows).

In contrast, an increase in fluid intake is associated with the excretion of a large volume of diluted urine to maintain the plasma concentration of osmotically effective solutes within a very narrow range. This is enabled by a decrease in AVP secretion. As a consequence, aquaporins do not relocate to cells’ luminal membrane of the collecting duct; the water permeability of the cells remains low, and water is not reabsorbed. This mechanism leads to the excretion of dilute urine with decreased osmolality (see Figure 1, green arrows).

### 1.3. Nephrolithiasis, Hydration Status and Urine Dilution

Nephrolithiasis is the only disorder that has been consistently associated with chronic low daily water intake and dehydration status [23]. In stone formers, there is strong evidence that hyperhydration resulting in urine dilution contributes to increased average interval for the recurrence of kidney stones but also recurrence rate [24]. Populations at risk of dehydration, such as anorexia nervosa patients [25,26], those with inflammatory bowel disease [27], Crohn’s disease patients [28] or even athletes exposed to acute dehydration [29], could exhibit a higher frequency of nephrolithiasis. Bar-David et al. also reported a higher rate of kidney stones in a hot environment [30].

A high intake of fluids, especially water, is still the most powerful and certainly the most economical means to prevent nephrolithiasis, and it is often not used to its advantage by stone formers [31]. The recently published study of the NHANES evaluates the association between water intake and hydration status with nephrolithiasis risk at the population level [32]. They showed positive dose–response associations of nephrolithiasis risk with insufficient hydration. Encouraging a daily water intake of more than 2.5 L/day and maintaining a urine output of 2 L/day is associated with a lower prevalence of nephrolithiasis. This issue is the same in pediatric populations. As shown by Miller and Stappelton, in a population of 50 children [33], urine volume is a risk factor for idiopathic calcium oxalate urolithiasis. Penido et al. also confirmed in a retrospective study with a population of 222 children that oliguria (defined as 24 h urine excretion below 1 mL/kg) is an etiologic factor for nephrolithiasis [34].

While specific measures, such as an increase in urine pH, can be applied for the treatment of uric acid or cystine stones, a 24 h urine volume greater than 2L is the main recommendation for common calcium oxalate stones, which account for more than 80% of all stones worldwide. Indeed, through the increase in fluid intake, urines are diluted, leading to decreased concentrations of lithogenic components in urine (mainly calcium and oxalate) and encouraging the expulsion of crystals thanks to the decrease in renal intratubular time transit (see Figure 2). Thus, urine dilution decreased calcium urine crystallization [35]. In stone former patients, a high recurrence rate strongly suggests that daily fluid intake remains insufficient. Up to 85% of all stone patients could lower their risk of stone recurrence with higher fluid intake [36]. In a prospective study with 181 adults with idiopathic calcium nephrolithiasis, it was shown that an increase in fluid intake to an average urinary volume of 1.7 L/day failed to prevent stone recurrence, while patients who successfully increased their daily urine volume up to 2.1 L/day on average did not make new stones [37]. Thus, urine dilution is the most effective tool to prevent calcium oxalate stone recurrence. In fact, the main driving force for calcium oxalate crystal formation is not the concentration of calcium or oxalate taken separately but the calcium oxalate molar product pCaOx. The frequency of calcium oxalate crystalluria increases with the increase in pCaOx. When pCaOx is greater than 4.5 (mmol/L)^2^, most urine samples contain calcium oxalate crystals [37]. Because doubling urine volume divides the molar product of calcium oxalate by four, it is the best therapeutic measure to decrease the frequency of CaOx crystals in urine and then to prevent stone recurrence [37]. All recommendations for kidney stone prevention suggest a daily urine volume between 2L [38,39] and 2.5L [40,41] per day. For cystine stone former patients, this daily urine volume should even be above 3L per day [42]. Patients with primary hyperoxaluria and preserved kidney function are advised to have hyperhydration, corresponding to 3.5–4 L/d in adults and 2–3 L/m^2^ body surface area in children to be consumed over 24 h [43].

Thus, avoiding dehydration and increasing fluid intake to dilute urine is key to preventing stone recurrence, and increasing diuresis is the main recommendation of physicians to prevent stone recurrence. To evaluate the adherence of nephrolithiasis patients to these recommendations, monitoring their hydration status through urine dilution is of great interest for personalized medicine. This monitoring of hydration status and of urine dilution requires different tools that have to be accessible both in the laboratory, during medical consultation, and/or at home in everyday life. Thus, the aim of this review is to summarize and discuss these different tools.

## 2. Which Kind of Urine Should Be Used for Hydration Status and Urine Dilution Monitoring?

In Europe, most centers perform analysis on both 24 h urine collection and spot urine. Urinary indices on spot urine (mainly osmolality and specific gravity) are not always correlated with 24 h urine indices [44]. Thus, they have to be interpreted in accordance with the type and time of urine collection.

### 2.1. Spot Urine

Spot urine indices depend on the time of collection. Urine collected in the morning before any food or fluid intake, so called fasting morning urine, is more concentrated than 24 h urine collection [2,45,46] because of the lack of fluid intake during the night period and urine accumulation in the bladder. When urine is collected after the bladder is properly voided before fluid intake, urine dilution may be observed as early as 30 to 60 min after a water load. Kovacs et al. even showed that nearly 3 h are required to normalize urine indices after acute dehydration, followed by fluid intake with a rapid increase in urinary output [47].

### 2.2. 24 h Urine Collection

Urine collection at 24 h is mostly considered the gold standard for urine hydration monitoring in daily life [48]. In recommendations, 24 h urine collection should be performed at least once a year at stone former patients’ follow-up [40]. Nonetheless, these collections are not so routinely performed in adults [49], as well as in children or adolescents [50,51], mostly because of the difficulty of the procedure. In this way, verification of the completeness of the collection has to be systematically checked. This collection must be performed in appropriate support, which yields easily quantified 24 h urine volume. These 24 h urine collections are mostly performed on plain urine, while several centers add acid or antibacterial agents to improve urine preservation for biochemical measurements [45]. In all cases, 24 h urinary creatinine excretion has to be measured to detect under or overcollection of urine; and avoid misinterpretation of results [52]. As 24 h urine collection reflects the whole day [48], they negate diurnal and nocturnal variations [53] and the consequences of daily activities [54]. Moreover, when punctually performed, they cannot reflect behavior throughout the day [55].

## 3. Laboratory Method

Monitoring hydration status ideally requires plasma osmolality measurement. Nonetheless, in nephrolithiasis patients, the most important factor that has to be controlled is urine concentration, especially dilution. Two main tools are widely used to assess urine dilution in laboratories: urine osmolality and urine-specific gravity (USG).

### 3.1. Urine Osmolality

Urine osmolality is the concentration of osmotic solutes present in the urine. It is measured using either a freezing point depression osmometer or, more rarely, a vapor pressure depression osmometer. This measure is highly reproducible with an analytical coefficient of variation lower than 0.4% [56]. Urine osmolality is highly stable in urine, up to 8 h at room temperature, 24 h at +4 °C, or even longer when stored at −20 °C [57]. Urine osmolality depends on two parameters: (1) the number of solutes and (2) the volume of fluid intake. Regarding the quantity of solutes, the most important ones are sodium, potassium and urea. In physiological conditions, their amounts depend mainly on diet because osmole elimination in urine is closely related to daily osmole intake and, in particular, daily protein intake [58], but also to calcium, magnesium and sugar consumption [59].

Regarding fluid intake, when it is insufficient, a small volume of highly concentrated urine is produced, leading to elevated urine osmolality; meanwhile, for a subject with a large fluid intake, a large amount of urine is produced, resulting in low urine osmolality. Urine osmolality physiologically ranges from 60 to 1200 mOsm/kg. Thus, intra-individual variation in urine osmolality is significant with a 28.3% variation coefficient, inter-individual variation is even higher, with a 57.9% variation coefficient [56]. Manz and Wentz showed that mean osmolality measured on 24 h urine collection varied from 360 mOsm/kg in Poland to 860 mOsm/kg in Germany, mainly because of the cultural differences in dietary fluid and osmole intake [60].

Urine osmolality measurement has many advantages. First, it is a non-invasive and cheap method. It is sensitive enough to detect small changes in hydration status and urine dilution. For 1-unit variation of plasma osmolality, there is a 100-unit variation in urine osmolality, showing a large deviation window [60]. Nonetheless, urine osmolality could also be elevated in the presence of glucose (diabetes patients) or urea (high protein diet), which have an osmotic effect, leading to a false conclusion regarding undiluted urine.

Urine osmolality measured on 24 h urine collection has been shown to predict calcium oxalate crystallization [61] with a high risk of kidney stone formation when 24 h urine osmolality is higher than 577 mOsm/kg in men and 501 mOsm/kg in women. These results are in accordance with several studies that suggest the use of a cut-off of 500 mOsm/kg as a reference for daily adequate fluid intake [62,63]. Kang et al. showed in a retrospective cohort of 5724 nephrolithiasis patients that a 24 h urine osmolality below 564 mOsm/kg was associated with longer stone-free recurrence [64].

Of note, estimation of urinary osmolality with machine-learning processes has been recently evaluated. Such methods include conductivity measurement, in addition to biochemical parameters assessment, but their performances are lower than those obtained with traditional osmolality measurements [65,66]. Moreover, to our knowledge, no study evaluates the use of urine calculated osmolality in nephrolithiasis patients’ prevention.

### 3.2. Urine-Specific Gravity

Urine-specific gravity (USG) corresponds to the measurement of urine density, defined as the weight of urine solution compared to that of an equal volume of distilled water. Thus, USG is affected by fluid intake and the particles in urine samples. The weight of plain water is equal to 1000 g/dm^3^, whereas normal urine samples usually range from 1013 to 1029 g/dm^3^. In physiological conditions, USG intra-individual variation is negligible with only a 0.4% variation coefficient. Inter-individual variation is also very low with a 1.0% variation coefficient [56]. In stone former patients, USG should be below 1010 to confirm optimal urine dilution to prevent recurrences [67].

USG can be directly measured via gravimetry by weighing aliquots of known volumes of urine on an analytical balance. These methods provide the most precise measurements of specific gravity; numerous gravimetric methods exist (e.g., falling drop, gravity beads, urinometer), but none of them are routinely used in laboratory conditions [68]. USG can also be measured indirectly with a refractometer that gives immediate results with low technical requirement. The refractive index is the ratio of velocity of light in air to the velocity of light in solution (in this case, urine). This change in velocity causes deviation (refraction) in the path of light. The degree of refraction is proportional to the number but also the type of particles, especially the chemical structure and the number of double bonds [68].

Finally, this method could be an alternative to urine osmolality measurement with good correlation between these two measurements, especially when this comparison is performed in clean urine samples [69,70,71]. Indeed, USG has one main disadvantage: it is also affected by the size and molecular weight of the particles dissolved in urine. USG can increase when unusual quantities of larger molecules, such as glucose, proteins, urea or even radiocontrast agents, are found in the urine, generating falsely elevated USG that suggests falsely highly concentrated urine [68]. Finally, its variations are lower than those of urine osmolality because it is affected by both the number and size of the particles, making USG less efficient than urine osmolality in estimate urine dilution.

### 3.3. Crystalluria

The main consequence of highly concentrated urine is the formation of crystals, which can be the first stage of stone recurrence in lithiasis patients. However, it is widely accepted that the presence of crystals is not a marker for stone formation. In fact, urine from healthy subjects may contain crystals without any pathological consequences. However, several papers have reported that the frequency of urine crystals was lower in controls than in stone formers [72,73]. Moreover, in a prospective study of stone recurrence in stone formers followed for an average of 6.7 years, crystalluria frequency was shown to be strongly correlated with the formation of new stones [37]. Specifically, 87.5% of patients who had crystals in more than 50% of their morning urine samples developed new stones, while only 15.6% of patients who did not have recurrence during follow-up had positive crystalluria in more than half of their urine samples. Thus, it may be clinically relevant to follow stone former patients over the long term through the study of crystalluria. Unfortunately, pre-analytical constraints, such as the time elapsed after voiding, can induce changes in urine crystals. In addition, analysis of urinary crystals is a time-consuming examination. As such, the study of crystalluria is a tool that cannot be performed in all laboratories. However, it helps to better understand the benefits of high fluid intake.

On the other hand, it is accepted that USG should be less than 1010 to prevent stone recurrence. Comparison between USG and the frequency of crystals in urine showed that less than 35% of urine samples with USG below 1011 contained crystals, while those with a USG value greater than 1011 contained crystals in more than 59% of cases (Figure 3) [37].

## 4. Home Method

At-home monitoring remains the most convenient way to assess urine dilution as it gives the patient rapid feedback on fluid intake and allows rapid correction if needed. These methods have to be cheap, easy to perform, not time consuming and should not require technical expertise.

Of note, body weight change, commonly found in the literature as a marker for global hydration monitoring, is not discussed here. If rapid body weight change is effectively linked to a gain or loss of water, it is mainly found in acute dehydration situation such as physical exercise or disease (diarrhea, etc.). Moreover, rapid body weight change shows a gain or loss of water without any indication about which compartment, extra or intracellular, is concerned and therefore cannot be related to urine dilution or, even more so, stone formation risks. Finally, monitoring precise body weight changes at home require accurate scale devices that are not widely available.

### 4.1. Urine Colour

One of the easiest indices to evaluate urine dilution outside laboratory measure is urine color [74], as first defined in 1994 by Armstrong et al. [69]. The color scale, composed of eight different tones from very pale yellow to brownish green (most diluted to most concentrated urine) has recently been studied to estimate urine osmolality in healthy adults. From a color index of 4 (i.e., yellowish green), the estimation of urine osmolality ≥ 500 mOsm/kg has good sensitivity (0.88) but poor specificity (0.64). Urine color can be affected by dietary factors, illness or medications [75], leading to important intra and inter-individual variability (30.9 and 47.4% variation coefficient, respectively) [56] and a poor correlation with urine osmolality [76]. However, these studies focused on 24 h urine collection as the reflection of urine dilution throughout the day, and this has never been investigated in patients with nephrolithiasis. In the meantime, it remains possible, although uneasy, for patients to monitor and report the color of their 24 h urine. Finally, seeing the color of their urine after each urination is still the easiest way to estimate, if urine is not pale enough, the need to increase fluid intake.

### 4.2. Urinary Strips for Urine-Specific Gravity

Urine-specific gravity (USG), as it is assessed by urinary strips (also used to monitor urine pH or glucosuria), is another tool that can be used at home by patients or by physicians in their offices. Khorami et al. found USG to be more efficient at stimulating water intake in nephrolithiasis patients than 24 h urine volume measurements [77]. Using a colorimetric assay, USG is estimated and could approximate urine osmolality, only if urine pH ranges between 7 and 7.5 [68]. Indeed, assessing urine density with strips in acidic pH tends to lead to a lower correlation with urine osmolality [78] because of falsely increased USG determination. Tables have been published to correct USG measurements with urinary strips according to urine pH values [79]. These defects could partially be corrected with the use of an automatic strip reader. Moreover, other modifications in urine composition, such as the presence of glucose, bilirubin, urobilinogen, protein or ketones, make urine density measurements by strips poorly correlated to urine osmolality [80]. A German study analyzing 340 first morning urine samples demonstrated reasonably good correlation between refractometry and single test strip results for USG estimation [81]; meanwhile, Rowat et al. showed a lack of accuracy between the refractometry measurement and urine color via urine test strip measures of USG in a study of 174 samples in 20 stroke patients [82]. Overall, the use of strips to measure USG as an estimation of urine dilution should not be recommended [83], especially as it has not been evaluated in patients with nephrolithiasis.

The different available tools to monitor urine dilution are summarized in Table 1 with their respective principles, advantages and disadvantages.

### 4.3. Fluid Intake Monitoring

If urine dilution can be difficult to monitor outside of the laboratory, fluid intake is easy to keep track of. The easiest way for the patient is to simply note the volume (half a bottle, a bottle, two bottles, etc.) of fluid they drink every day, as it correlates well with the daily volume of urine excreted in nephrolithiasis patients [84]. With the rise of smart devices (smartphones, smartwatches, etc.) and the need for better compliance to high water intake, new technologies have been developed to monitor fluid intake and hydration status. As a matter of fact, patients seem to be seeking for such connected devices that could help them increase their water consumption [85].

An initial study was published in 1991 using the Urimho device, yielding an assessment of urine-specific gravity and urine conductivity according to a color scale [86]. The device greatly correlated with laboratory measures and helped patients for a better compliance. In the last five years, a new kind of device has been developed: smart bottles. By accurately measuring the volume drank from them [87,88] and sending reminders to drink via a smartphone app, smart bottles successfully increased water intake in nephrolithiasis patients [89]. Moreover, these devices facilitate tracking fluid intake as volume consumed, which is transmitted and stored on the patient’s smartphone and can easily be accessible for physicians.

When patients were asked what kind of connected devices would be most likely to encourage them to increase their fluid intake, a wristband or bracelet was the most cited item [85]. This is what has been recently studied by Rodin et al., who tested a body monitor hydration device paired with a smartwatch [90]. In 240 healthy adults, a photoplethysmographic sensor whose optical properties change when different molecules are detected in sweat (sodium, potassium, water, etc.). After physical activities, changes detected by this equipment were strongly correlated with body mass change, the only hydration marker assessed by the authors. If such devices seem promising in term of practicality, their utility and their efficiency in improving water intake in nephrolithiasis patients remains to be clarified.

## 5. Conclusions

Hyperhydration aiming to dilute urine is the cornerstone of the preventive medical treatment in nephrolithiasis patients, whatever the stone composition. Consequently, monitoring urine dilution in stone former patients is key to preventing stone recurrence and to help patients adhere to preventive treatment, especially beverage intake recommendations. If laboratory tools, and especially urine osmolality measurement, are the most informative indices, they are rarely performed in daily life and even rarely used in clinical trials. Thus, home monitoring seems the most convenient and efficient way for rapid feedback and correction, even though these tools are less sensitive and specific. Eventually, connected devices may be promising tools to prevent stone recurrence in nephrolithiasis patients.

## Figures and Tables

**Figure 1 nutrients-15-01642-f001:**
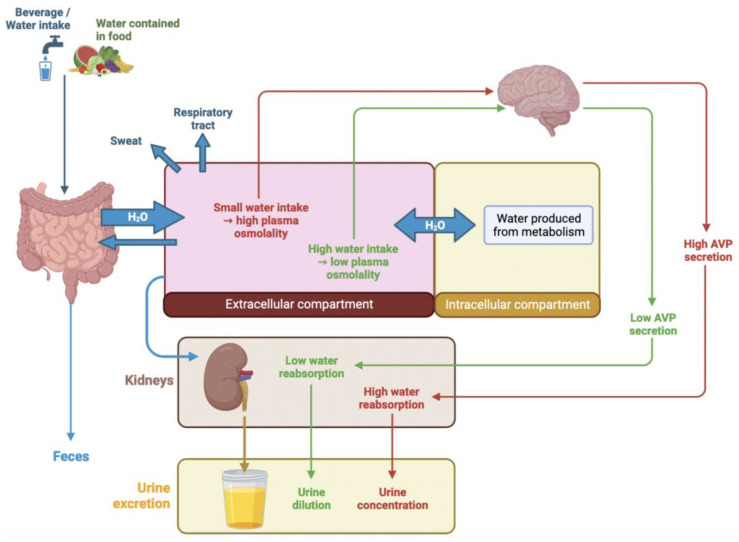
Physiological regulation of water balance. Fluid intake is mostly excreted by kidneys in response to water balance and AVP release. In the case of limited water intake (red arrows), intracellular volume decreased, leading to high AVP release and thus maximum water reabsorption by the kidney. Finally, a small amount of concentrated urine is excreted. On the other hand, high fluid intake (green arrows) leads to increased intracellular volume and thus to suppression of AVP release, and finally an increased amount of diluted urine.

**Figure 2 nutrients-15-01642-f002:**
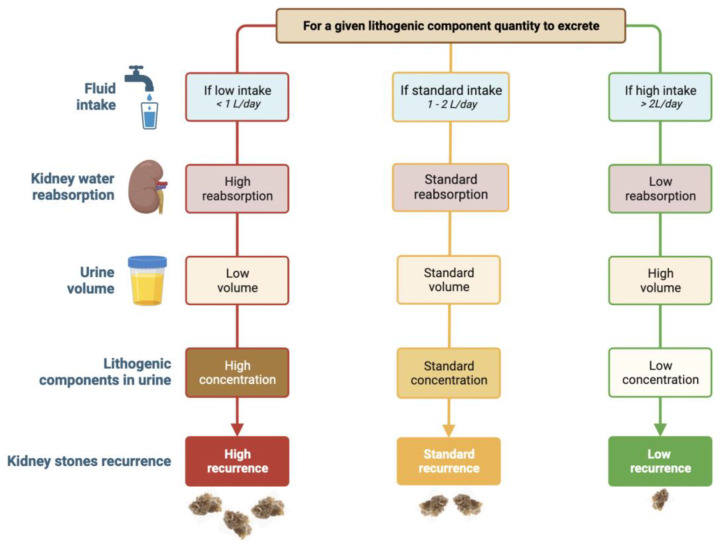
Relationship between fluid intake and kidney stone recurrence. For a given lithogenic component quantity to excrete, the volume of fluid intake is responsible for the amount of kidney water reabsorption and thus final urine volume, lithogenic component concentration in urine and, finally, kidney stone recurrence. With low fluid intake (left part), kidney water reabsorption is high, leading to a low volume of urine with a high concentration of lithogenic components, thus favoring kidney stone recurrence. With high fluid intake (right part), kidney water reabsorption is low, leading to a high volume of urine with low concentration of lithogenic components, preventing stone recurrence.

**Figure 3 nutrients-15-01642-f003:**
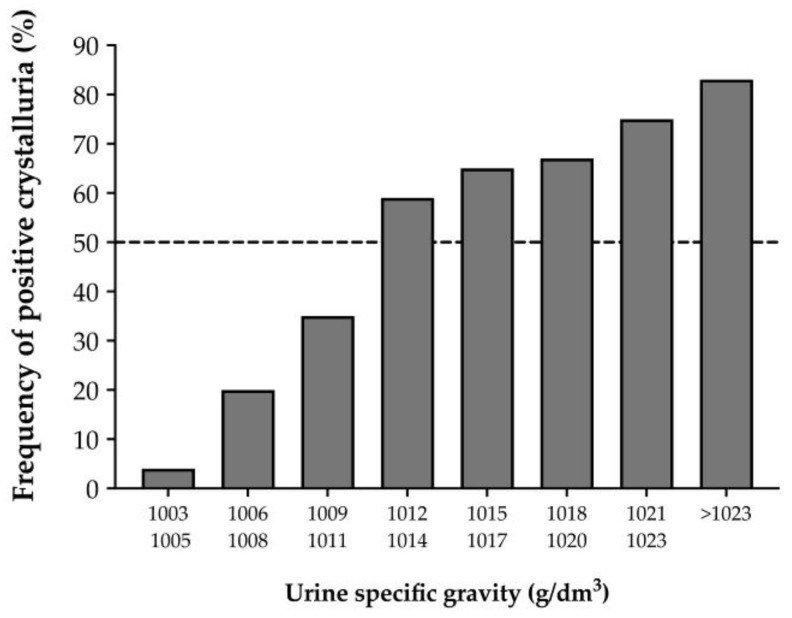
Relationship between urine-specific gravity and frequency of positive crystalluria (based on [37]).

**Table 1 nutrients-15-01642-t001:** Summary of principles, advantages and disadvantages of monitoring tools.

	Principle	Advantages	Disadvantages
Urine osmolality	Freezing point depression (or more rarely vapor pressure depression)	− Cheap − High Sensitivity	- Only in labs - Misinterpretation in cases of glucosuria or high protein intake
Urine-specific gravity	Refractometry	− Mild sensitivity	- Only in labs - Misinterpretation with glucosuria, high protein intake, radiocontrast agents
	Urine strip with automatic strip reader	− In physicians examination room	- Moderate specificity
	Urine strip	− At home	- Interferences when urine sample pH does not range between 7 and 7.5 - Poor sensitivity - Poor specificity
Crystalluria	Miscroscopic detection of crystals in urine	− High sensitivity − Highly predictive	- Available only in few specific labs - Time consuming - Important pre-analytical contraints
Urine color	Visual examination	− At home − Cheap	- Poor sensitivity - Poor specificity

## Data Availability

Not applicable.

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
