# Peer review of "How to Monitor Hydration Status and Urine Dilution in Patients with Nephrolithiasis"

_nutrients, 2023, doi:10.3390/nu15071642_

Round 1

Reviewer 1 Report

The review is interesting and tackles a little-discussed topic. I approve the publication.

Author Response

We thank reviewer 1 for his/her comment

Reviewer 2 Report

In this manuscript, the authors describe the mechanism of urine dilution and methods for monitoring to analyze the best methods for monitoring hydration status and urine dilution in patients with nephrolithiasis. This study is very interesting, but there are some deficiencies that need to be revised.  

1. When introducing the regulation of hydration balance and Nephrolithiasi, hydration status and urine dilution, etc., the author should use some figures to express this part more intuitively and vividly.

2. In line 122, What is the specific function of urine volume? In the study of . Penido et al., how did they classify the amount of urine output?

3. Line 277, What population is this study based on? What is the sample size? What are the corresponding statistical results?

4. Authors should summarize the principles, advantages and disadvantages of different monitoring tools in a table.

Author Response

In this manuscript, the authors describe the mechanism of urine dilution and methods for monitoring to analyze the best methods for monitoring hydration status and urine dilution in patients with nephrolithiasis. This study is very interesting, but there are some deficiencies that need to be revised.  

We thank reviewer 2 for his/her comments on our manuscript.

  1. When introducing the regulation of hydration balance and Nephrolithiasis, hydration status and urine dilution, etc., the author should use some figures to express this part more intuitively and vividly.

As suggested by reviewer 2, the addition of figures would help readers. In this way we added a first figure (figure 1) to describe physiological process of water balance, and urine dilution. This figure has been added at the end of paragraph 1.1 (line 71). This figure is introduced in sentence line 61: “Regarding fluid outputs, loss of water mainly occurs via kidney urine excretion, via sweating, through the respiratory tract and via the feces, as described in figure 1.”. Then, the figure is cited line 102 “Finally, any acute or chronic dehydration, even moderate or mild, is responsible for an increase in kidney water reabsorption, a decrease in urine volume and an increase in the different solutes’ concentrations (see figure 1, red arrows).” and line 108 “As a consequence, aquaporins do not relocate to cells luminal membrane of the collecting duct: the water permeability of the cells remains low and water is not reabsorbed. This mechanism leads to the excretion of dilute urine with a decreased osmolality (see figure 1, green arrows.”

The corresponding additional legends is: “Figure 1. Physiological regulation of water balance” with corresponding description beginning line 73: “Fluid intakes are mostly excreted by kidneys in response to water balance and AVP release. In case of small water intake (red arrows), intracellular volume decreased leading to high AVP re-lease and thus maximum water reabsorption by the kidney. Finally, a small amount of concentrated urine is excreted. On the other way, a high fluid intake (green arrows) leads to an increase intracellular volume, thus to suppression of AVP release, and finally an increase amount of diluted urine.”

We then added a second figure to describe the relationship between lithogenic components concentration in urine and fluid intake. This figure has been added at the end of paragraph 1.3 of the manuscript, line 161. This figure aims at describing the relationship between low, standard or high fluid intakes, and their consequences on the different steps leading to stone recurrence or not. This figure is first introduced line 138: “Indeed, through the increase in fluid intake, urines are diluted leading to decreased concentrations of lithogenic components in urine (mainly calcium and oxalate) and en-courage the expulsion of crystals thanks to the decrease in renal intratubular time transit (see figure 2).”

The corresponding additional legends is:

“Figure 2. Relationship between fluid intake and kidney stone recurrence”.

The corresponding description of the figure is as follow line 165:

“For a given lithogenic component quantity to excrete, the volume of fluid intake is responsible for the amount of kidney water reabsorption, thus final urine volume, lithogenic component concentration in urine and finally kidney stone recurrence. With low fluid intakes (left part), kidney water reabsorption is high, leading to a low volume of urine with high concentration of lithogenic components, thus favoring kidney stone recurrence. With high fluid intakes (right part), kidney water reabsorption is low, leading to a high volume of urine with low concentration of lithogenic components, preventing stone recurrence.”

  1. In line 122, What is the specific function of urine volume? In the study of . Penido et al., how did they classify the amount of urine output

We thank reviewer 2 for this comment. Indeed, in Penido et al. study, oliguria, is one of the most common etiologies of pediatric primary urolithiasis. In children, oliguria is defined as a 24-hour urine excretion below 1m mL/kg/day. In manuscript, this sentence beginning line 122 (now line 132) was rephrased as below:

“Penido et al. also confirm in a retrospective study with a population of 222 children that oliguria (defined as a 24-hour urine excretion below 1 mL/kg) is an etiologic factor for nephrolithiasis [34].“

  1. Line 277, What population is this study based on? What is the sample size? What are the corresponding statistical results?

We thank reviewer 2 for this remark. Indeed, this study is the same as previously described line 289 “Moreover, in a prospective study of stone recurrence in stone formers followed for an average of 6.7 years, crystalluria frequency was shown to be strongly correlated with the formation of new stones [37].”

This study is based on the follow up of 181 calcium nephrolithiasis patients for a mean duration of 6.7 years. Thus, the sentence initially line 277 has been deleted, but the description of this study has been completed in sentence line 291:  

Specifically, 87.5% of patients who had crystals in more than 50% of their morning urine samples developed new stones; “while only 15.6% of patients who did not have recurrence during follow up had positive crystalluria in more than half of their urine samples.”

  1. Authors should summarize the principles, advantages and disadvantages of different monitoring tools in a table.

We thank reviewer 2 for this suggestion, and we added a table 1 to the manuscript. This table is cited at the end of paragraph 4.2 and is introduced with sentence line 357:

“The different available tools to monitor urine dilution are summarized in table 1 with their respective principles, advantages and disadvantages.”

Reviewer 3 Report

In this manuscript, authors thoroughly reviewed the roles of laboratory and home methods to monitor hydration status in patients with nephrolithiasis. The subject of this review seems to be unique and interesting for many readers. There are two points that authors should clarify in the revised manuscript, as follows.

1. Many guidelines for reducing the recurrence of nephrolithiasis recommend sufficient hydration such as 2 o 3 L/day of water intake. Do authors consider that such hydration is insufficient in not a few patients with nephrolithiasis, and that thus, introduction of monitoring methods is needed in clinical practice? If authors consider that water intake suggested in guidelines is inappropriate, evidence should be presented. As the reviewer’s recommendation, monitoring should be suggested for personalized medicine but not for alternative to measurement of water intake.

2.  Although authors suggested some methods to estimate urine dilution, are there any methods with robust evidence to reduce development or recurrence of nephrolithiasis, among them?

Author Response

In this manuscript, authors thoroughly reviewed the roles of laboratory and home methods to monitor hydration status in patients with nephrolithiasis. The subject of this review seems to be unique and interesting for many readers. There are two points that authors should clarify in the revised manuscript, as follows.

We thank reviewer 3 for his.her comments on our manuscript.

  1. Many guidelines for reducing the recurrence of nephrolithiasis recommend sufficient hydration such as 2 o 3 L/day of water intake. Do authors consider that such hydration is insufficient in not a few patients with nephrolithiasis, and that thus, introduction of monitoring methods is needed in clinical practice? If authors consider that water intake suggested in guidelines is inappropriate, evidence should be presented. As the reviewer’s recommendation, monitoring should be suggested for personalized medicine but not for alternative to measurement of water intake. 

As underlined by reviewer 3, numerous guidelines recommend a water intake between 2 and 3 L/day in recurrence prevention of most common cases of nephrolithiasis. In some specific cases such as hyperoxaluria or cystinuria patients, recommendations even suggest water intake above 3L/day for cystinuria patients (Servais, et al. Cystinuria: clinical practice recommendation. Kidney Int 2021, 99, 48-58,) and above 3.5L/day for primary oxaluria patients (Groothoff, J.W.; et al. Clinical practice recommendations for primary hyperoxaluria: an expert consensus statement from ERKNet and OxalEurope. Nat Rev Nephrol 2023, 19, 194-211). Nonetheless, patients’ adherence to these recommendations is not always complete. In such cases, kidney stone recurrence is more frequent than for patients who fully adhere to fluid intake guidelines. In this way, monitoring of urine dilution is indeed an additional tool to prevent kidney stone recurrence, in association with fluid intake recommendations.

In this way, the sentence line 175 has been added:

“To evaluate the adherence of nephrolithiasis patients to these recommendations, monitoring of their hydration status through urine dilution is of great interest for personalized medicine.”

This point is now also underlined in the conclusion line 392.

Consequently, monitoring urine dilution in stone former patients is a key point to prevent stone recurrence and to help patients adhere to the preventive treatment, “especially beverage intake recommendations.”

  1. Although authors suggested some methods to estimate urine dilution, are there any methods with robust evidence to reduce development or recurrence of nephrolithiasis, among them?

We thank reviewer 3 for this comment. Indeed, very few studies use monitoring urine dilution tools such as measurement of urinary osmolality or USG in randomized clinical trials (RCT). Beverage intake estimation and 24-hour urine volume recording are the most common parameters using in those RCT. Nonetheless, one important retrospective study performed by Ho Won Kang et al. was based on the exploration of 5724 first-nephrolithiasis patients (Kang, H.W.; et al.Twenty-four-hour urine osmolality as a representative index of adequate hydration and a predictor of recurrence in patients with urolithiasis. Int Urol Nephrol 2019). Patients were followed between 1994 and 2017. This study showed a longer stone recurrence-free period in patients for whom 24-hour urine osmolality was below 564 mOsm/kg than those with U Osm above 564 mOsm/kg. Thus, the results of this study are now commented in the manuscript. The following sentence has been added line 247:

“Kang et al. showed in a retrospective cohort of 5724 nephrolithiasis patients that a 24-hour urine osmolality below 564 mOsm/kg was associated with a longer stone free recurrence [65].”

In the same way, in the conclusion, the sentence line 393 has been rephrased:

“If laboratory tools, and especially urine osmolality measurement, are the most informative indices, they are rarely performed in daily life and even rarely used in clinical trials.”
